# Research on the Current Application Status of Magnesium Metal Stents in Human Luminal Cavities

**DOI:** 10.3390/jfb14090462

**Published:** 2023-09-08

**Authors:** Xiang Chen, Yan Xia, Sheng Shen, Chunyan Wang, Rui Zan, Han Yu, Shi Yang, Xiaohong Zheng, Jiankang Yang, Tao Suo, Yaqi Gu, Xiaonong Zhang

**Affiliations:** 1School of Medicine, Anhui University of Science and Technology, Huainan 232000, China; chenxiang0913@outlook.com; 2School of Stomatology, Anhui Medical College, Hefei 230601, China; xiayan@ahyz.edu.cn; 3Department of Biliary Surgery, Zhongshan Hospital, Fudan University, Shanghai 200032, China; shen.sheng@zs-hospital.sh.cn (S.S.); ruizan@fudan.edu.cn (R.Z.); suo.tao@zs-hospital.sh.cn (T.S.); 4Shanghai Engineering Research Center of Biliary Tract Minimal Invasive Surgery and Materials, Shanghai 200032, China; zswangcy@sina.cn; 5Department of General Surgery, Shanghai Xuhui Central Hospital, Shanghai 200031, China; 6State Key Laboratory of Metal Matrix Composites, School of Materials Science and Engineering, Shanghai Jiao Tong University, Shanghai 200240, China; lipervan@sjtu.edu.cn (H.Y.); yangsh2017@sjtu.edu.cn (S.Y.); 7Department of Hepatopancreatobiliary Surgery, Huainan Xinhua Hospital Affiliated to Anhui University of Science and Technology, Huainan 232000, China; zxh7643323@163.com (X.Z.); yjk13955459952@163.com (J.Y.)

**Keywords:** Mg, Mg-based stent, biodegradable stents, luminal cavities applications, clinical translation

## Abstract

The human body comprises various tubular structures that have essential functions in different bodily systems. These structures are responsible for transporting food, liquids, waste, and other substances throughout the body. However, factors such as inflammation, tumors, stones, infections, or the accumulation of substances can lead to the narrowing or blockage of these tubular structures, which can impair the normal function of the corresponding organs or tissues. To address luminal obstructions, stenting is a commonly used treatment. However, to minimize complications associated with the long-term implantation of permanent stents, there is an increasing demand for biodegradable stents (BDS). Magnesium (Mg) metal is an exceptional choice for creating BDS due to its degradability, good mechanical properties, and biocompatibility. Currently, the Magmaris^®^ coronary stents and UNITY-B^TM^ biliary stent have obtained Conformité Européene (CE) certification. Moreover, there are several other types of stents undergoing research and development as well as clinical trials. In this review, we discuss the required degradation cycle and the specific properties (anti-inflammatory effect, antibacterial effect, etc.) of BDS in different lumen areas based on the biocompatibility and degradability of currently available magnesium-based scaffolds. We also offer potential insights into the future development of BDS.

## 1. Introduction

The human body comprises a diverse range of organs that play a crucial role in maintaining regular physiological functions [1]. These organs can be broadly categorized into two groups based on their structure: parenchymal organs and hollow organs. Parenchymal organs refer to organs that have parenchymal tissues, such as the heart, liver, and lungs, which are usually composed of dense tissues. Hollow organs, on the other hand, refer to organs that have hollow spaces and are able to accommodate other substances. For example, organs like the blood vessels, intestines, and bile ducts belong to the category of hollow organs, as they have hollow structures that can contain food, liquids, waste, and other substances [2]. When the lumen of hollow organs is damaged, infected, or obstructed by tumors, it can lead to a reduction in luminal patency, further resulting in luminal obstruction and causing varying degrees of organ damage [3]. In severe cases, it can lead to organ necrosis and even death.

The implantation of stents is widely regarded as the most effective method for treating the stenosis of hollow organs in clinical practice [4]. Stents, as commonly used intracorporeal implants, perform an essential function in supporting and restoring luminal patency. Figure 1 illustrates that commonly used stents in clinical practice encompass a wide range according to the service environment, such as coronary stents [5], vascular stents [6], biliary stents [7], tracheal stents [8], esophageal stents [9], intestinal stents [10], and urethral stents [11]. Each location of the lumen has specific requirements for stents, but regardless of the location, good mechanical properties and biocompatibility are fundamental prerequisites. Conventional stent systems encompass (Table 1) both non-degradable organic and metal materials. The organic stents are typically fabricated from various materials such as polyethylene (PE), polyurethane (PU), and polytetrafluoroethylene (PTFE) and are typically used in treating benign luminal stenosis conditions. The plastic stents are more cost-effective and easier to implant; however, the strength and stiffness of plastic stents are generally lower, which limits the diameter of the stent, resulting in frequent stent obstructions necessitating replacement. Metallic stents are predominantly composed of stainless steel and nickel–titanium alloys. They are generally deployed for luminal obstructive conditions demanding extended support (beyond 6 months). Metal stents possess a wider lumen and superior patency; however, they come at a higher cost and are predisposed to luminal adhesion, making their removal considerably more challenging once obstruction occurs. During long-term implantation, permanent material stents can potentially result in complications such as displacement, re-obstruction, and inflammation. Additionally, the removal of these stents often necessitates a second surgery, which can impose additional financial and psychological burdens on the patient.

Consequently, research into biodegradable stents has assumed paramount importance. Such stents can gradually degrade and be absorbed, circumventing the issues associated with long-term retention and secondary surgical removal, thus presenting significant potential in stent selection and research direction. Biodegradable materials are classified into two categories: biodegradable organic materials and biodegradable metallic materials. Biodegradable organic materials include polylactic acid (PLA), polycaprolactone (PCL), and others. These materials exhibit good biocompatibility in the surrounding tissues following implantation. Nonetheless, in general, the mechanical properties of absorbable polymers are insufficient to meet the radial strength required for luminal support, necessitating an increase in the strut thickness of the stent to enhance its support capabilities. Furthermore, their applicability within the lumen is constrained, and they are prone to scaffold displacement, mirroring the characteristics of plastic scaffolds. The biodegradable metallic materials encompass magnesium, iron, and zinc metals. This paper, in particular, emphasizes biodegradable magnesium. Owing to its impressive mechanical strength, magnesium-based stents feature relatively thin strut thicknesses, resulting in larger inner diameters and an expanded range of applications. Additionally, magnesium’s degradation product is Mg^2+^, a trace element within the body known for its exceptional biocompatibility. However, the manufacturing of magnesium-based scaffolds comes at a higher cost and exhibits rapid degradation; therefore, surface modification of magnesium-based scaffolds is imperative to prolong their degradation cycle. However, some retrospective studies have shown that these permanent implants can cause adverse reactions, including tissue hyperplasia, inflammation, and adhesion during long-term treatment, which require a second surgery to remove or replace the stent [12,13], increasing medical costs and surgical risks. Therefore, there is an increasing demand for the next generation of innovative biodegradable stents, which fully degrade within a certain period to maintain luminal support with good biocompatibility as well as good bioactive ability. Among the biodegradable materials, magnesium (Mg) metal is considered an ideal material for bioresorbable stents due to its degradability, good mechanical properties, and biocompatibility [14,15,16].

Mg is an essential element in the body [17,18]. According to the recommendations of the National Institutes of Health (NIH) in the United States, adults need to consume over 300 mg of Mg ions per day to be involved in over 300 enzyme-catalyzed reactions, maintaining membrane stability, energy metabolism [19], protein synthesis, DNA replication, and cell skeleton activation. The mechanical properties and density of Mg metals are similar to those of human skeletal tissue, meeting the mechanical requirements for supporting the human body effectively, reducing foreign body sensation, and improving biomechanics. Extensive in vivo and clinical studies have shown that Mg implants react with the body fluids and completely degrade in the body. Excess Mg ions can be excreted through urine and feces, avoiding the potential for biotoxicity and allergic reactions during the implantation process. Interestingly, Mg implantations and their degradation products have demonstrated a range of biological functions, such as promoting bone formation, inhibiting inflammation, and exhibiting anti-tumor properties [20,21]. To date, eight Mg-based medical devices have been certified, including Magmaris^®^ coronary stents and UNITY-B^TM^ biliary stents [22,23,24]. The development and application of various types of Mg stents present significant opportunities but also pose several challenges. As previously discussed, different types of hollow organs have distinct mechanical, degradation, and biological activity requirements for stents due to the variations in cavity diameter, fluid environment, and surrounding tissue composition. Consequently, this review aims to outline the performance criteria, current research progress, and future trends in the field of stents, encompassing vascular, coronary, biliary, tracheal, esophagus, urethral, and intestinal stents.

## 2. Types of Mg-Based Supports

### 2.1. Vascular Stents

#### 2.1.1. Cardiovascular Stents

The vascular system in the human body is primarily composed of arteries, veins, and capillaries, which are extensively distributed throughout the human body. Its primary role is to facilitate the transportation of oxygen, transfer nutrients, and eliminate metabolic waste [25]. The cardiovascular system, a vital subsystem of the vascular system, is responsible for circulating oxygenated blood to all parts of the body. When blood vessels narrow, it can result in cardiovascular disease. Since the invention of coronary stents in 1980 [26], they have emerged as an effective method for treating cardiovascular diseases [27]. Vascular scaffolds mainly come into contact with the blood, and the pH value of blood is generally maintained between 7.35 and 7.45, being slightly alkaline (Figure 2). Initially, vascular stents were predominantly made from non-degradable inert materials [28]. However, once implanted in the human body, they function as a permanent foreign body, constantly stimulating the inner wall of the blood vessels and triggering inflammation [29,30,31]. In theory, stents can achieve cardiovascular remodeling and restore their normal physiological function after a period of 6 months following implantation. Therefore, for biodegradable cardiovascular stents, the mechanical support duration of the stents needs to exceed 6 months. The Mg alloy, as one of the ideal biodegradable metal materials, has a degradation cycle that essentially meets the clinical requirements [32]. Additionally, the Mg alloy exhibits good anti-platelet adhesion and a low probability of thrombus re-formation [33,34]. These characteristics make it the ideal biomaterial for the preparation of vascular scaffolds.

#### 2.1.2. Coronary Stents

Coronary atherosclerotic heart disease, also known as “coronary heart disease”, occurs when atherosclerotic lesions develop in the coronary arteries, causing the narrowing or blockage of the blood vessels, which leads to myocardial ischemia, hypoxia, or necrosis and results in heart disease. With the aging population and lifestyle changes, coronary heart disease is one of the most common cardiovascular diseases worldwide. Millions of people are diagnosed with coronary heart disease each year, including different types such as angina and myocardial infarction. Since Andreas Gruntzig first performed percutaneous transluminal coronary angioplasty (PCI) in 1977, PCI technology has gradually evolved and its indications have expanded [35]. In 1986, Puel and Sigwart successfully implanted the first coronary stent in the human body [26]. Nowadays, the implantation of coronary stents has become a milestone in coronary interventional therapy to reduce the occurrence of restenosis and stenosis stenting. They work in narrowed coronary arteries to restore blood flow and alleviate myocardial ischemic symptoms. In recent decades, there has been an increasing demand for coronary stents. In 2003, drug-eluting stents (DES) were introduced in clinical applications. These stents are loaded with drugs such as sirolimus, everolimus, and zotarolimus, which are primarily immunosuppressive and antiproliferative drugs [36]. By inhibiting neoplastic endothelial proliferation, DES significantly reduces the incidence of stent restenosis and the need for re-intervention [37]. DES releases drugs locally at the site of the vascular injury, achieving effective drug concentration over a specific period while minimizing systemic drug toxicity. However, DES implantation requires dual antiplatelet therapy, with a lifelong anticoagulant intake thereafter [38,39,40]. Late in-stent thrombosis remains a concern [41]. A large number of studies have indicated that permanent coronary stents alter hemodynamics, reduce vascular response, and limit coronary vessels’ diastolic force [42,43]. In addition, some side effects, such as thrombosis, long-term endothelial stimulation, endothelial dysfunction, and local chronic inflammatory response have been performed. To address these issues, a biodegradable coronary stent, especially Mg alloy stents, represents an optimal alternative, which offers adequate mechanical properties, completely absorbs within a specific timeframe, and exhibits bioactive ability. The Mg alloy coronary stent promotes vascular healing, restores normal vascular function, reduces platelet adhesion, prevents thrombosis, and serves as a drug carrier without severe inflammatory reactions during degradation [44,45].

Heublein et al. [46] fabricated coating on the AZ21 (Mg-Al-Zn-Mn) Mg alloy coronary stent to enhance corrosion property, which completes the degradation within 56 days in vivo, with the stent structure already being destroyed within 35 days. HE staining revealed significant endothelial hyperplasia at the site of the stent implantation, indicating a vascular inflammation reaction induced by the degradation products of the Mg metal stent. Ron Waksman et al. [47] prepared a bioabsorbable WE43 (Mg-Y-RE-Zr, exhibiting high strength and hardness and exceptional corrosion resistance, but it comes with a high manufacturing cost and relatively complex processing) Mg alloy stent, which also required 56 days for complete in vivo degradation. Similar to the AZ21 Mg alloy, the implantation of the Mg alloy vascular stent in animal blood vessels showed a loss of stent continuity and integrity within 28 days. However, compared to the control group with stainless steel stents, the Mg alloy stent reduced endothelial damage and inflammatory response. After 3 months of placement, the Mg stent demonstrated a significantly smaller area of stenosis compared to the stainless-steel stent group, which is attributed to Mg’s inhibition of endothelial proliferation. Upon complete degradation of the Mg alloy stents, it was observed that the area of the vessels with Mg alloy stents was significantly improved compared to the structure-destroyed stents at 28 days. This proved that the long-term (>3 months) implantation of Mg alloy stents could promote coronary vascular remodeling. Thus, the requirements for Mg alloys in coronary stent implantation include a prolonged degradation period of at least >3 months, good biocompatibility, and reduced stimulation of endothelial damage. While drug-eluting stents (DES) significantly reduce the incidence of restenosis compared to bare stents, they still have limitations, such as permanent vessel stimulation and late thrombosis [48,49]. Therefore, drug-eluting bioresorbable scaffolds (BRS) have emerged as an alternative to DES. Zhu et al. [50] developed a novel Mg-based scaffold (Figure 3) using a newly patented Mg-Nd-Zn-Zr alloy (JDBM). The scaffold’s surface was coated with rapamycin-coated poly (D, L-lactic acid) (PDLLA/RAPA), which extended the degradation cycle of JDBM. Animal experiments demonstrated that the degradation cycle of JDBM BRS scaffolds exceeded 6 months, with partial degradation observed at 3 months of implantation while maintaining the main structure intact. At 6 months, degradation intensified but the scaffold still maintained intact mechanical properties. These results confirmed the effective prolongation of the JDBM BRS degradation cycle via the PDLLA/RAPA coating. Cellular experiments also indicated that the coating reduced epithelial–smooth muscle cell proliferation and mitigated inflammatory stimulation caused by the stent on the vessel.

In the comparison of various Mg alloy coronary stents presented in Table 2, it is evident that the current requirements for biodegradable coronary stents include a degradation cycle of >6 months, a maintenance of mechanical strength, good biocompatibility, and a reduction in damage to the vascular endothelium. Mg alloys and their degradation products can partially reduce endothelial proliferation, and bioresorbable Mg stents exhibit negative charges during degradation [51], which may possess potential antithrombotic properties. Studies have also demonstrated that Mg can attenuate ischemia–reperfusion injury [52], and its inhibition of endothelin-1 production can prevent endothelin-mediated vasoconstriction [53,54]. Bioresorbable Mg scaffolds in the field of coronary stents are still undergoing advancements. These scaffolds not only exhibit visible benefits in mitigating coronary inflammatory response and late thrombosis but also hold potential for loaded drug coatings that further reduce in-stent restenosis and prolong the lifespan of Mg-based scaffolds. These findings underscore the suitability of the Mg alloy as a promising material for medical applications.
Figure 3The application of magnesium-based coronary stents in human and animal models. (**A**,**B**): The experiment of Magmaris^®^ implanted into human coronary arteries in vivo (Figure **A** illustrates the end-to-end implantation technique of multiple Magmaris® stents, where: 1 denotes the initial Magmaris^®^ stent, 2 signifies the tantalum marker on the first Magmaris^®^ stent, 3 indicates the balloon associated with the second Magmaris^®^ stent, 4 represents the tantalum marker on the second Magmaris^®^ stent, and 5 corresponds to the second Magmaris^®^ stent itself) [22,55]; (**C**): the experiment of Magmaris^®^ implanted into pig coronary arteries in vivo (The yellow circles depict the strut framework of the coronary stent, with Figure 2 in particular showcasing a clearer view of the stent struts.) [56]; and (**D**,**E**): the experiment of JDBM implanted into rabbit coronary arteries in vivo [50].
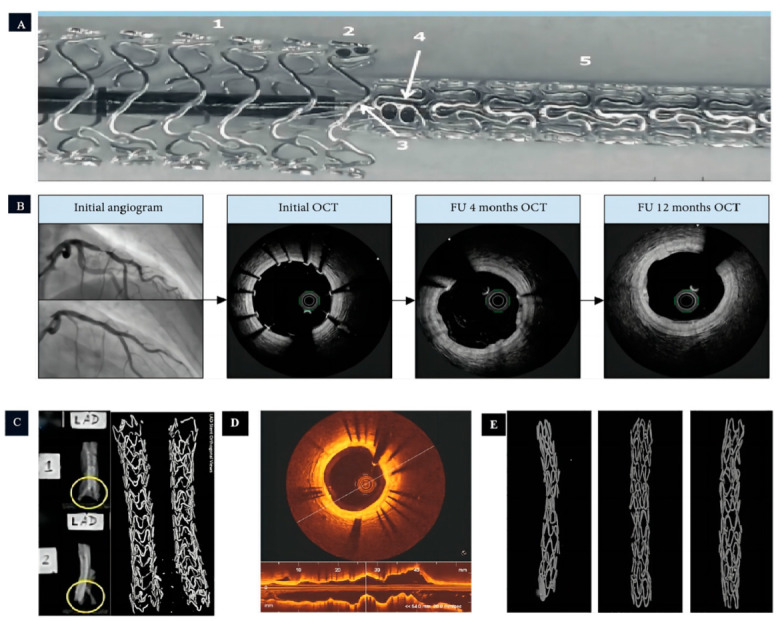



#### 2.1.3. Other Vascular Stents

The occlusive vascular disease remains one of the leading causes of death in humans. Vascular stents are inserted into diseased segments of blood vessels to provide support, reduce vascular elasticity retraction, promote vascular remodeling, and maintain unobstructed blood flow. Currently, vascular stents are primarily categorized into different types, including coronary stents, cerebral vascular stents, renal artery stents, aortic artery stents, and others.

While stent implantation is an effective treatment for occlusive vascular disease, autologous vein bypass grafting remains a significant therapeutic option [57]. However, approximately 50% experiences vein graft failure (VGF) [58] and lose functionality within 10 years, primarily due to vein graft stenosis. Current endovascular treatments include balloon dilatation, bare metal stents (BMS), and drug-eluting stents (DES), all of which have shown effectiveness in improving the prognosis of VGF [59,60,61,62]. Similar to coronary stents, VGF treatments pose long-term issues such as thrombosis and endothelial hyperplasia. Bioabsorbable Mg scaffolds (BMASs) offer new hope for VGF treatment. Li et al. [63] conducted a study using AZ31 (Mg-Al-Zn-Mn, with high machinability, low density, and low corrosion resistance) Mg alloy scaffolds implanted into the carotid artery and abdominal aorta of experimental rabbits (Figure 4). The degradation cycle of the AZ31 Mg alloy was approximately 4 months, with structural damage to the AZ31 Mg alloy scaffolds observed at 2–3 months. Furthermore, a comparison between the AZ31 group and the control stainless-steel group revealed that the implantation of the AZ31 Mg alloy promoted an early-stage lumen diameter increase and vascular endothelialization.

The prevalence of cerebral aneurysms ranges from 1% to 7%, making it one of the most common cerebrovascular diseases [64,65,66,67]. Coil embolization is the primary treatment for cerebral aneurysms [68,69]. However, this method has certain disadvantages, including a relatively low rate of complete occlusion, a high rate of recanalization, high treatment costs, and limited anatomical cures [70,71]. In recent years, vascular stenting has emerged as an alternative approach to isolate aneurysms [72]. However, most of the available stents are permanent metal stents that can induce intimal hyperplasia and lead to restenosis. In the case of cerebral aneurysms, biodegradable Mg stents have become an alternative treatment option. Cui et al. [73] established an animal model of common carotid artery collateral aneurysms (CCA) and implanted Mg alloy overlay stents (Macs) into the carotid arteries of experimental rabbits. Most of the struts in the Macs stent degraded after 3 months of implantation, and by 6 months of implantation, the Macs stent was completely degraded. During the degradation process of the Macs stent, endothelial cells were observed on the remaining struts, and the rest of the stent was fully endothelialized. Compared to the control group with the Willis Coated Stent (WCS), the Macs stent showed a faster endothelialization process and a better sealing of CCA collateral aneurysms.

Based on the different clinical requirements and implantation sites, the requirements for biodegradable vascular scaffolds may vary. Like biodegradable coronary stents, the general requirements include a degradation cycle of >6 months, a maintenance of mechanical strength, and good biocompatibility. However, there is an additional requirement specific to certain clinical needs, which is the promotion of endothelialization in blood vessels. For instance, in procedures such as autologous venous bypass grafting and coil embolization of aneurysms, the implanted biodegradable vascular scaffolds are expected to facilitate the rapid endothelialization of the vessel and ensure the integrity of the lumen, thus improving therapeutic outcomes.

### 2.2. Biliary Stents

The bile duct is a long tubular tissue responsible for transporting bile and is one of the digestive organs. Bile has a pH that is maintained between 7.6 and 8.6 (Figure 2) due to its high concentration of alkaline substances such as bile salts, cholesterol, and bilirubin. The liver and gallbladder play a significant role in regulating bile pH. The liver regulates bile pH primarily by secreting bile acids during bile synthesis, while the gallbladder is responsible for storing and concentrating bile and has a minimal impact on bile Ph. Bile duct stenosis typically occurs as a result of biliary stones, tumors, inflammation, or intraoperative bile duct injuries [24]. It is generally categorized into benign biliary strictures (BBS) and malignant biliary strictures (MBS). BBS is mainly caused by biliary tract injuries and gallstones, while MBS is primarily caused by biliary tract tumors. BBS and MBS have different requirements for biliary stents. Currently, biliary stents that are commonly used in clinical practice can be classified into plastic stents and metal stents [74]. Plastic stents are typically used for patients with benign biliary obstruction and has a lifespan of less than 3 months due to their tendency to displace and cause re-obstruction, necessitating frequent replacements. In contrast, metal stents have superior durability and are suitable for use for 3 months or longer. However, removing metal stents can be challenging due to lumen adhesion, and they are also more costly, placing a greater financial burden on patients. Although the specific requirements for biliary stents vary depending on the type of bile duct stenosis, in general, a secondary operation is necessary to remove biliary stents. Biodegradable stents can degrade within the body, reducing the necessity for a subsequent surgery and alleviating any discomfort experienced by the patients. However, biliary stents have strict requirements for degradation due to the complex composition of bile fluids in the service environment, which accelerates the mechanical deterioration of the stents. Although biodegradable Mg biliary stents prevent complications associated with prolonged stent placement in the bile duct, corrosion resistance remains a huge challenge in a clinical trial.

Currently, there are several types of Mg alloy biliary stents available, such as Mg- 2Zn, Mg-6Zn, AZ31, JDBM, etc. [25,74]. As different magnesium alloys incorporate distinct alloying elements, their characteristics including strength, hardness, and density exhibit variations. In general, the pH levels within the human body’s lumens tend to be mostly neutral, with some being mildly acidic or slightly alkaline. Within a neutral environment, JDBM demonstrates superior biocompatibility and resistance to corrosion. Likewise, WE43 demonstrates analogous traits in such conditions, though it is often not the primary therapeutic choice due to its higher cost. Conversely, AZ21 and AZ31 alloys display a slightly lower resistance to corrosion and often necessitate additional coatings to enhance their degradation cycle. In conclusion, the selection of magnesium alloys suitable for various luminal scaffolds necessitates a comprehensive evaluation considering factors such as specific application requirements, desired biocompatibility, and degradation cycle. Guo et al. [75] included a bare Mg alloy stent and a coated stent. An immersion experiment in vitro indicated the JDBM bare stent suffered structural damage and lost its supportive function after 12 weeks of immersion. In contrast, the JDBM-MgF_2_/PDLLA-coated stent retained its basic structure after 12 weeks of immersion. In the in vivo experiments, the overall structure of both the JDBM bare stent and the JDBM-MgF_2_/PDLLA-coated stent disappeared after 30 days of implantation. After 60 days of implantation, both groups of stents completely degraded, and the bile ducts showed a smooth endothelium without stenosis or fibroplasia. Hao et al. [76] prepared AZ31B Mg alloy stents via the surface micro-arc oxidation method. After implantation into the bile ducts, the AZ31B stents began to undergo structural damage after 2 months and completely degraded after 3 months. Liu et al. [77] prepared an AZ31 Mg alloy stent (Figure 5), which, after 3 months of implantation in the biliary tract, exhibited destruction of the stent structure. After 6 months, the stent completely degraded with only a small amount of metal residue, demonstrating good corrosion resistance of the AZ31 Mg alloy stent. It has successfully manufactured an MZ2 biliary scaffold and investigated the stratification of degradation products in this scaffold. This was accomplished via the examination of its in vitro degradation process in the bile and HBSS solution, as well as its in vivo degradation within the bile duct. These studies have established a solid groundwork for the utilization of degradable magnesium scaffolds in the field of biliary ducts [78]. The development of the Mg alloy as a new biodegradable material in the field of biliary stents attracts more attention. The good biocompatibility and mechanical properties of the Mg alloy make it an ideal material for biliary stents. However, the rapid corrosion rate of the Mg alloy in the bile duct environment hinders its suitability for the desired service life of biliary stents. Prolonging the corrosion resistance of the Mg alloy remains a key focus of biliary stent research. Nevertheless, there is a broad prospect for the use of the Mg alloy in the field of biliary stents.

### 2.3. Tracheal Stents

The microenvironment of the pH value in the trachea is relatively alkaline (from 7.3 to 7.6) but fluctuates in certain respiratory diseases, such as chronic obstructive pulmonary disease (COPD) and airway infections. Tracheal stenting is used to treat diseases that cause airway obstruction, such as congenital airway stenosis (CTS) and acquired airway stenosis. These conditions can make it difficult to breathe and cause shortness of breath. [80]. Airway obstruction can be further classified as either auto-causal (due to the lack of tracheal structure) or iatrogenic (resulting from disruptions to the tracheal structure) [81,82]. The primary goal of treating patients with tracheal stenosis is to maintain airway patency and stability [83]. Surgery remains associated with a high intraoperative mortality rate and postoperative complications, making it a less favorable option. Airway stenting can rapidly reconstruct the airway and alleviate symptoms such as respiratory distress [84,85], making it a potentially effective method. However, conventional stents may not meet the rapid growth needs of infants with CTS. Hence, the development of a new generation of biodegradable medical implants, such as the Mg alloy, has been explored to reduce the incidence of associated complications.

Xue et al. [86] conducted experiments using Mg-Zn-Ca alloys and JDBM Mg alloys to process degradable Mg scaffolds. According to the results in vitro and in vivo, the JDBM tracheal stent maintained its structure for over 2 months, underwent uniform degradation, and showed no apparent inflammatory reaction. The trachea exhibited no significant morphological abnormalities. These experiments confirmed that the Mg alloy tracheal stents did not induce severe inflammatory reactions in the airway and were harmless to vital organs, confirming the feasibility of biodegradable Mg alloy tracheal stents used in tracheal stenosis. Therefore, for degradable tracheal stents, in addition to possessing good mechanical properties and complete degradation within a certain period (>2 months), they also need to exhibit good biocompatibility and reduce irritation to the tracheal epithelium, thus minimizing the airway’s inflammatory response.

### 2.4. Esophageal Stents

The esophagus exhibits slight variations in pH levels across its different segments. The Upper Esophageal Sphincter (UES) has a neutral pH of 6.5–7.5. The Esophageal Body (EB) has an acidic pH of 6.0–7.0. The Lower Esophageal Sphincter (LES) is more acidic, with a pH of 5.5–6.5 (Figure 2). The requirements for stent implantation in different parts of the esophagus are generally similar. The main condition that necessitates esophageal stents is esophageal stenosis, particularly in cases of benign esophageal stenosis caused by conditions such as gastro-esophageal reflux, esophageal erosion, and injuries, which are common in clinical practice [87]. The goal of treatment is to alleviate dysphagia, and esophagectomy dilatation is a commonly used clinical procedure that often requires repeated sessions [88,89]. Therefore, stenting has emerged as an alternative method for treating benign esophageal stenosis [90]. However, permanent metal stents can disrupt the physiological structure of the esophagus and are typically removed after a period of more than 3 months, where complications occur such as damage to the esophageal mucosa, esophageal perforation, bleeding, and infection after the removal of stents [91]. Consequently, there is an urgent need to research and develop biodegradable esophageal stents.

Mg is more prone to degradation in an acidic environment, which avoids the long-term retention of the stents. Therefore, some biodegradable Mg esophageal stents with polymer coating were designed to prolong the anti-corrosion ability. Liu et al. [92] prepared the stent coated with poly (lactic acid)-hydroxy acetic acid copolymer (PLGA) incorporating the antiproliferative agent paclitaxel (PTX), which showed relative biosafety after being implanted in rabbits. The evaluation of stent displacement within the first 3 weeks revealed a displacement rate of 58.3% (7/12), but the esophagus of all rabbits remained patent. Yuan et al. [93] developed an AZ31 magnesium alloy stent coated with PCL-PTMC (Figure 6). In in vitro degradation experiments, the PCL-PTMC-coated AZ31 esophageal stent was tested in a neutral solution, showing a 10% reduction in stent mass after 4 weeks. After 10 weeks of testing, the stent’s residual weight was 65% of its original weight. However, in an acidic environment, the stent completely degraded after 10 weeks. Among the 10 rabbits implanted with PCL-PTMC-coated AZ31 esophageal stents, two exhibited displacements. HE staining revealed reduced esophageal wall remodeling and no significant inflammatory response. These findings suggest that the PCL-PTMC-coated AZ31 esophageal stent exhibits outstanding biocompatibility. The experimental group showed a decrease in the thickness of the esophageal wall, a reduced inflammatory response, and an increased lumen area, which prevented the growth of granulation tissue. Therefore, the requirements for biodegradable esophageal stents include good mechanical properties, complete degradation within a specific period (>3 weeks), good biocompatibility, and reduced irritation to the esophageal epithelium in order to prevent stent displacement.

### 2.5. Urethral Stents

The urethra has an acidic pH ranging from 5.0 to 7.0, as shown in Figure 2. In urology, stents were first introduced in the 1970s, with silicone and polyurethane being the preferred materials for ureteral stents due to their ability to reduce encrustation [94]. The current first-line treatment plan requires follow-up cystoscopy and stent removal after a certain time, which can cause discomfort to patients, including lumbar and abdominal pain, urinary tract irritation, infections, and stone formation. Failure to remove the ureteral stent promptly after treatment may lead to complications such as renal failure [95,96,97], and multiple anesthesia procedures can impose physical and economic burdens on patients [98,99,100]. Currently, double-J stents (DJ) are commonly used in clinical practice as they have been shown to reduce complications, facilitate stent removal, and minimize patient discomfort [101]. However, the ideal ureteral stent should be completely biodegradable in the body without causing cytotoxicity [102]. Natural biodegradable materials for ureteral stents include alginate, gelatin, and hyaluronic acid, while synthetic polymer materials include polylactic acid, polyglycolic acid, and polylactic acid–glycolic acid. In recent years, Mg and Mg alloys have also been explored as potential materials for biodegradable ureteral stents. Studies have demonstrated that Mg, Mg-Y alloys, and AZ31 alloys can degrade in artificial urine and exhibit antimicrobial activity, highlighting the potential of Mg alloys for urological applications (Figure 7) [103,104]. Therefore, the antimicrobial activity of Mg alloys serves as an advantage for their use as biodegradable materials.

Tie et al. [105] conducted a study using ZJ41 (Mg-Zn-Sr, high strength, high hardness, good machinability, and low density) Mg alloy semi-solid rheological scaffolds [106]. These scaffolds exhibited superior mechanical properties compared to general Mg-Zn alloys. In the experiment, the ZL41 Mg scaffolds began to exhibit structural damage after 7 weeks of implantation and completely degraded at 12 weeks. Due to the antimicrobial activity of the ZJ41 alloy, the frequency of urination in the experimental animals was significantly lower two weeks after surgery. After 6 weeks, the urinary frequency returned to normal levels. Furthermore, it not only validates the efficacy of degradable Mg stents as ureteral stents but also showcases the antimicrobial properties of Mg alloys, which can help alleviate adverse reactions associated with stent implantation. Therefore, a requirement for degradable ureteral stents is to possess certain antibacterial activity, which can inhibit biofilm formation and bacterial growth.

### 2.6. Intestinal Stents

The intestines are tubular structures in the digestive tract that extend from the stomach to the rectum. The pH values in different parts of the intestines may vary. For example, the jejunum typically has a slightly acidic pH ranging from 6.0 to 7.4, while the colon has a slightly acidic pH ranging from 5.5 to 7.0. Intestinal stenting is often required for the treatment of intestinal stenosis. Benign intestinal stenosis can be caused by surgical injuries, radiotherapy, or the effects of medications. Malignant intestinal stenosis is associated with intestinal malignant tumors or metastases. Both types of stenosis can greatly reduce a patient’s quality of life and may lead to complications such as malnutrition and weight loss. Conventional treatments often require repeated procedures. As an alternative, self-expanding metal stents have been used to treat benign intestinal stenosis. These stents provide long-term mechanical support to maintain intestinal patency. However, the implantation of metal stents can cause tissue hyperplasia, which can lead to serious complications such as perforation and intestinal obstruction [107]. To overcome these issues, biodegradable intestinal stents have been developed as an alternative to plastic or metal stents. These stents avoid the need for secondary surgery and reduce the risk of complications [108]. Currently, degradable materials used for intestinal stents include lactic acid, glycolic acid, caprolactone, Mg-based alloys, and iron-based alloys [109]. Mg alloys, in particular, have shown promise due to their good mechanical properties and biocompatibility. Studies have demonstrated that Mg alloys undergo degradation without significant inflammation or necrosis, and the degradation products do not exhibit significant toxicity to important organs. Therefore, degradable Mg alloy stents could serve as a suitable alternative to traditional intestinal stents.

Wang et al. [110] performed a study where three different Mg-Zn-Y-Nd alloy (WE42, boasts high strength, lightweight characteristics, exceptional corrosion resistance, and exclusive machinability. Nevertheless, its production cost is relatively high, options for its use are somewhat limited, and it is susceptible to humid environments) (Figure 8) scaffolds were prepared. Animal experiments revealed that PLLA/paclitaxel JDBM alloy scaffolds exhibited rapid degradation starting from the 8th day, complete degradation in 9–14 days, and showed a significant inhibitory effect on the growth of intestinal endothelial tissue. This inhibitory effect effectively suppressed the excessive growth of local intestinal tissues.

## 3. Clinical Applications of Mg-Based Stents

The Mg alloy, one of the most promising biodegradable materials, has a rich history of clinical applications. Its earliest recorded use dates back to 1878 when Edward ligated pure Mg to control bleeding, although it degraded rapidly. With the continuous development of society, Mg alloys have transformed and found applications in various fields (Table 3). Notably, they have been successfully utilized in the development of luminal stents, such as coronary stents and bile duct stents. The first version of a biodegradable metal stent (AMS-1) designed for human coronary arteries contained 93% Mg and 7% rare earth elements [111]. AMS-1 was a bare-metal stent, and subsequent improvements led to the development of AMS-2, which employed refined alloy compositions. To inhibit smooth muscle cell growth, a poly (lactic acid)-hydroxy acetic acid-coating-loaded paclitaxel was added to the surface, resulting in AMS 3 (DREAMS-1G) [112]. Further enhancements led to DREAMS-2G [113], which incorporated a 7 μm sirolimus-eluting poly-L-lactic acid coating on the surface, extending the degradation cycle to 12 months. In 2016, this stent was released as Magmaris^®^ (Figure 9) with a CE marking in the Euro, marking the first biodegradable drug-eluting metallic stent (Figure 5). Additionally, the UNITY-B^TM^ [79] biodegradable implant was developed as a complementary biodegradable metallic stent to AMG’s ARCHIMEDES biodegradable pancreatic-biliary stent (Figure 9). The UNITY-B^TM^ stent offers rapid, intermediate, and long-term degradation profiles. The ARCHIMEDES stent received CE mark approval in 2018 and was subsequently released globally via a partnership with Medtronic.

## 4. Physiological Functions of Mg

Magnesium metal implants have been extensively studied for their potential therapeutic applications due to their various biological activities. These implants have demonstrated anti-tumor [115,116], antibacterial [117], anti-inflammatory [118], and tissue healing promotion effects [119]. 

One notable characteristic of magnesium implants is their ability to undergo a hydrogen absorption reaction with body fluids. This reaction results in the release of magnesium hydroxide (Mg (OH)_2_) and hydrogen gas (H_2_) [120].

The degradation behavior of Mg implants can be controlled via external stimuli such as light and magnetism, enabling a controlled inhibitory effect on tumor growth. The released H_2_ from Mg implants has shown promising results in selectively inducing apoptosis (cell death) in different types of tumor cells, including gallbladder cancer, colorectal cancer, liver cancer, and more [121,122,123]. We previously found that magnesium degradation products inhibit various types of tumors, such as osteosarcoma tumors [124], ovarian cancer [115], colorectal cancer, and gallbladder cancers [21]. Specifically, hydrogen suppresses tumor cells by activating the P53-mediated lysosomal–mitochondrial apoptosis signaling pathway. Furthermore, another research found that Mg implants activate immune response [125,126], acting as a potential option for targeted cancer therapy.

Hydrogen gas is used as a therapeutic medical gas for treating some diseases, attributing to the reduction in highly cytotoxic reactive oxygen species (ROS), such as peroxynitrite (ONOO-) and hydroxyl radical (•OH) [127,128], without disrupting the signaling of normal cells [129]. Furthermore, we conducted an observation using a live cell microscope and found that the presence of hydrogen led to the generation of an apoptotic body in tumor cells. However, in normal cells, hydrogen could be metabolized without impacting the growth state of the cells. Similarly, when magnesium was implanted in the body, it was observed that hydrogen induced apoptosis in tumor tissues while causing minimal damage to the normal organs [122].

Mg implants can achieve antibacterial effects using different approaches. These include incorporating antibacterial elements in the implant alloy design, modifying the surface topography, and loading antimicrobial drugs [130]. These strategies help prevent bacterial colonization on the implant surface and reduce the risk of infection [131,132,133]. 

In addition to these properties, Mg implants promote tissue healing via collagen synthesis, angiogenesis, cell proliferation [134], and anti-inflammatory effects [92,135]. These mechanisms contribute to the overall tissue regeneration process.

Overall, the unique biological activities of Mg metal implants, such as their controlled degradation, selective tumor cell apoptosis, anti-inflammatory effects [136], and antibacterial properties, make them promising candidates for a wide range of medical applications in fields such as tubular lumen intervention and tissue engineering.

## 5. Conclusions

In summary, this article elucidates the application, translation and prospective development of biodegradable magnesium (Mg) alloys in the domain of medical stents. The investigation underscores the significance of corrosion resistance and biocompatibility in diverse anatomical cavities. The quintessential biodegradable Mg stent ought to possess stellar mechanical properties, biodegradability, and commendable biocompatibility while taking into account site-specific requisites. The utilization of degradable stents obviates the necessity for secondary interventions and proffers unparalleled advantages in disparate luminal environments. Mg alloys have garnered attention as exceedingly promising materials for multifaceted human implants, owing to their mechanical attributes and degradation kinetics. Nevertheless, challenges such as mechanical support duration, degradation kinetics, biological functionality, and compatibility with minimally invasive therapeutic modalities necessitate redressal for prospective device translation. Further investigation is imperative to fully explore the potential of biodegradable Mg alloy stents in specific domains. All in all, Mg alloys exhibit promise as biodegradable materials in the purview of medical stents.

## Figures and Tables

**Figure 1 jfb-14-00462-f001:**
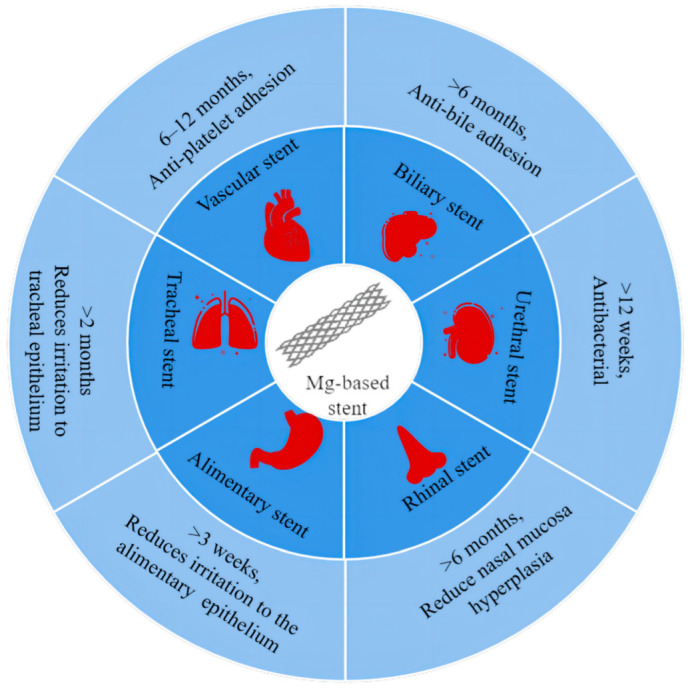
Degradation cycles and special requirements of magnesium-based stents in different cavities.

**Figure 2 jfb-14-00462-f002:**
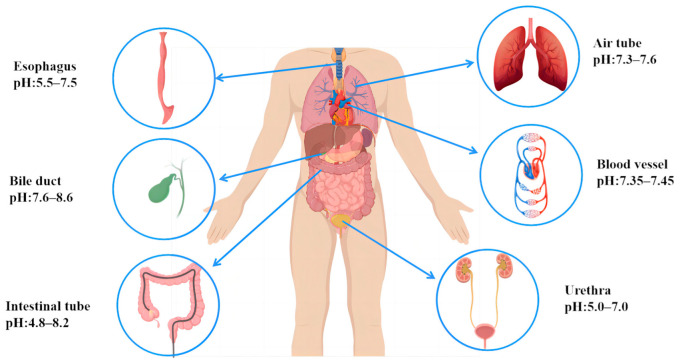
Body fluids and pH in different body cavities.

**Figure 4 jfb-14-00462-f004:**
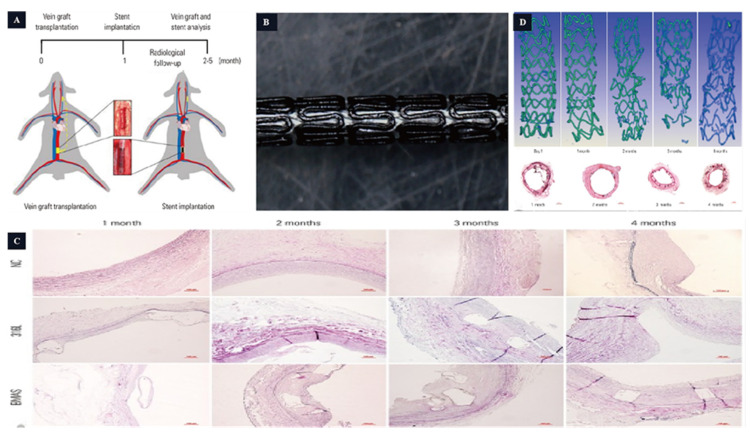
The magnesium-based scaffold is used in the vascular stent. (**A**,**B**): Stent design and animal model. (**C**): The grafted vein Verhoeff–Van Gieson staining. (**D**): The BMAS degradation process [63].

**Figure 5 jfb-14-00462-f005:**
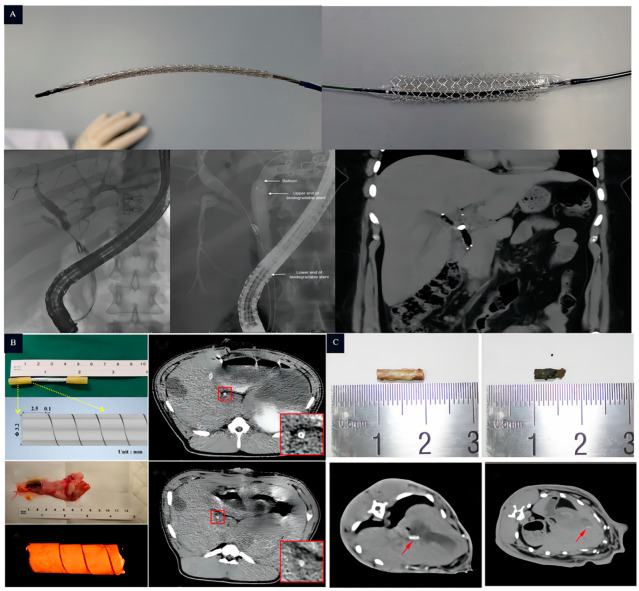
The application of magnesium-based biliary stents in human and animal models: (**A**): in vivo experiment of UNITY-B^TM^ biliary stent implanted into human bile duct (From top to bottom, the white arrows indicate the balloon, followed by the upper end and the lower end of the biodegradable stent) [79]; (**B**): the experiment of MZ2 implanted into pig bile duct in vivo (The arrows in the figure indicate the MZ2 biliary stent and its design, while the white highlighted area inside the red box provides a magnified view of the biliary stent 14 days after implantation) [78]; and (**C**): the experiment of AZ31 implanted into rabbit bile duct in vivo (The white highlighted shadow, indicated by the arrow, represents the AZ31 stent that has been implanted in the rabbit’s bile duct) [77].

**Figure 6 jfb-14-00462-f006:**
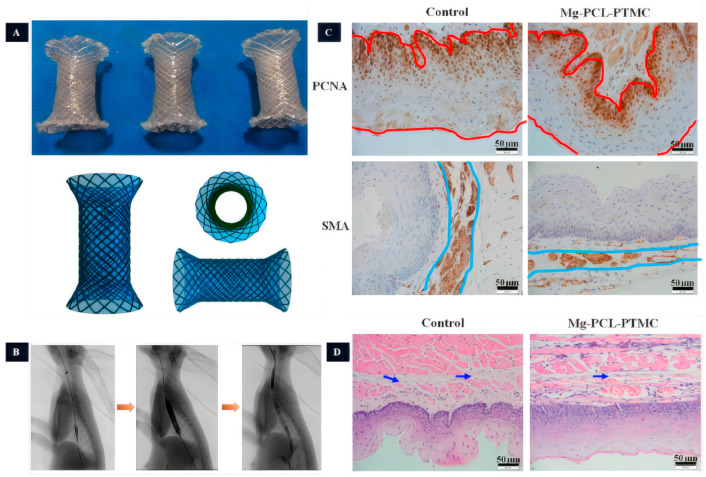
The study of Mg-based esophageal stents. (**A**): The PCL-PTMC coated Mg-stent shape. (**B**): Esophageal radiography after stent implantation (the yellow arrows indicate the imaging of the esophagus at different stages of stent implantation, including pre-implantation imaging, imaging after stent placement, imaging after balloon dilatation following stent placement, and assessment of esophageal patency after stent placement, respectively). (**C**): The magnesium stent group inhibits epithelial and smooth muscle cells (The red line represents the thickness of the epithelial layer, while the blue line represents the thickness of the smooth muscle layer). (**D**): Tissue HE staining (The arrows indicate the presence of collagen deposition in the submucosal layer.) [93].

**Figure 7 jfb-14-00462-f007:**
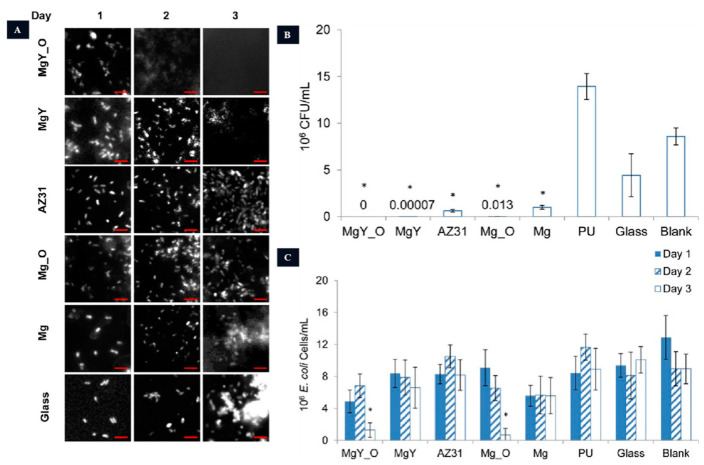
The anti-bacterial activity of Mg-based urethral stents. (**A**): Adhesion of *E. coli* on different samples. (**B**): Quantification of colonies after 16 h of incubation (* *p* < 0.05 compared to PU, glass, and the blank group). (**C**): Bacterial enumeration in AU solution following incubation with various materials (* *p* < 0.05 in comparison to PU, glass, and the control group on day 3) [104].

**Figure 8 jfb-14-00462-f008:**
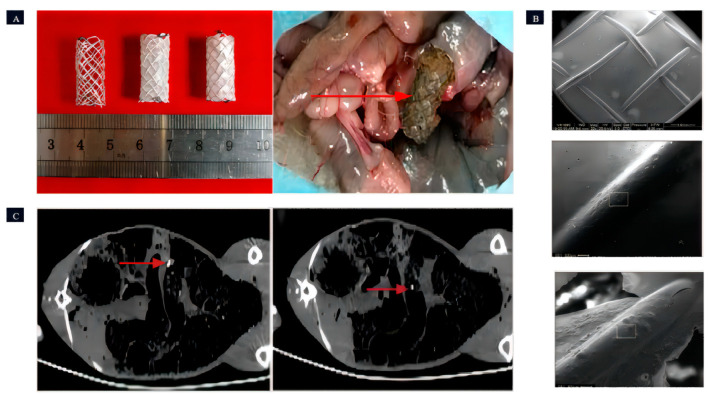
The Mg-based intestinal stents. (**A**): Coating status of intestinal stents: uncoated, PPLA-coated, and MAO/PLLA/paclitaxel-coated; surgical perspective of stent retrieval (the red arrow indicates the intestinal stent that was removed from the rabbit’s intestine). (**B**): Presents SEM images of the intestinal scaffolds with PLLA coating taken before implantation, 5 days post-implantation, and 8 days post-implantation, respectively (in the figure, the surface of the intestinal scaffold is visibly coated with a thin layer of PLLA. The PLLA coatings on both the unimplanted intestinal scaffold and the intestinal scaffold after 5 days of implantation were notably smooth and devoid of any cracks. However, after 8 days of implantation, cracks started to emerge in the PLLA coating of the intestinal scaffold). (**C**): CT images depict the front and rear extremities of the intestinal stent within a rabbit (the bright area indicated by the red arrow is the Nitinol ring that serves as the identifier for the stent) [110].

**Figure 9 jfb-14-00462-f009:**
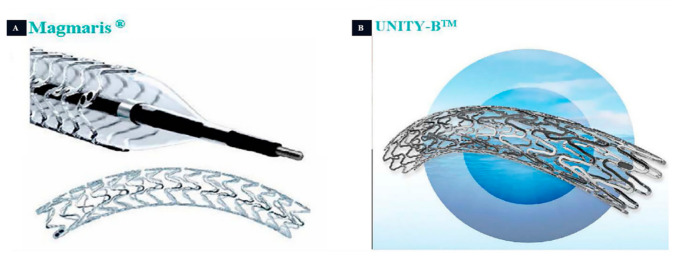
Conversion products of magnesium-based stents. (**A**): Magmaris^®^ coronary stent produced by Biotronik company; and (**B**): UNITY-B^TM.^ biliary stent produced by Q3 Medical Devices Limited [114].

**Table 1 jfb-14-00462-t001:** Comparison of conventional and biodegradable stents.

Classifications	Materials	Advantages	Disadvantages	Brand
Non-biodegradable organic stent	Polyethylene (PE) Polyurethane (PU) Polytetrafluoroethylene (PTFE)	Low cost Simple operation	Low strength Vulnerability to obstruction Ease of displacement Short life cycle (3–6 months)	Percuflex™/Biliary stent Zaontz/Ureteral stent Firlit-Kluge/Ureteral stent
Non-biodegradable metal stent	Stainless steel Nickel–titanium alloys	Long life cycle (>6 months) Less prone to in-stent obstruction	High costs Difficulty in secondary removal	EUROLIMUS™/Coronary stent Rontis Medical-Abrax™/Coronary stent EndoMAXX^®^/Oesophageal stent
Biodegradable organic stents	Polycaprolactone (PCL) Polylactic acid (PLA)	No need for a second surgery Good biocompatibility Elasticity and flexibility	Uncontrollable degradation rate Risk of displacement and obstruction Mechanical performance limitation	ARCHIMEDES™/Biliary stent Igaki–Tamai/Coronary stent ReZolve/Coronary stent
Biodegradable metal stents	Magnesium (Mg) Iron (Fe) Zinc (Zn)	No need for a second surgery Superior mechanical properties for a wide range of applications. Good biocompatibility.	High cost Uncontrollable degrada-tion rate	Magmaris^®^/Coronary stent UNITY-B™/Biliary stent

The stents listed in the table represent only a selection, not the complete set.

**Table 2 jfb-14-00462-t002:** Summary of degradable Mg alloy stents.

Material	Composition	Experimental Animal	Parenting Type	Implant Part	Time of Stent Integrity Failure (d)	Degradation Cycle (d)
AZ21	Mg-Al-Zn-Mn	Pig	Coronary stent	Coronary artery	35	56
WE43	Mg-Y-RE-Zr	Pig	Coronary stent	Coronary artery	28	56
JDBM	Mg-Nd-Zn-Zr	Pig	Coronary stent	Coronary artery	90	180
AZ31	Mg-Al-Zn-Mn	Rabbit	Intravascular stent	Carotid artery	60–90	120
JDBM	Mg-Nd-Zn-Zr	Rabbit	Intravascular stent	Carotid artery	90	180
JDBM	Mg-Nd-Zn-Zr	Dog	Biliary stent	Biliary tract	30	60
AZ31B	Mg-Al-Zn-Mn	Dog	Biliary stent	Biliary tract	60	90
AZ31	Mg-Al-Zn-Mn	Rabbit	Biliary stent	Biliary tract	90	180
JDBM	Mg-Nd-Zn-Zr	Rabbit	Tracheal stent	Trachea	/	60
JDBM	Mg-Nd-Zn-Zr	Rabbit	Esophageal stent	Esophagus	/	84
AZ31	Mg-Al-Zn-Mn	Rabbit	Esophageal stent	Esophagus	/	>70
ZJ41	Mg-Zn-Sr	Pig	Urethral stent	Urethra	49	84
WE42	Mg-Zn-Y-Nd	Rabbit	Intestinal stent	Intestinum tenue	8	14

**Table 3 jfb-14-00462-t003:** The applications of Mg-based implants in the clinical trial.

Product	Material	Alloy Composition	Coating	Coating Thickness (μm)	Drug	Application Area	Strut Thickness (μm)	Stent Design
AMS-1	WE31	Mg-Nd-Zn-Zr	/		/	Cardiovascular	80 × 165	4-crown 4-link
AMS-2	WE31	Mg-Nd-Zn-Zr	/		/	Cardiovascular	130 × 120	6-crown 3-link
AMS-3 (DREAMS-1G)	WE31	Mg-Nd-Zn-Zr	PLGA	1	Paclitaxel (0.07 μg/mm)	Cardiovascular	130 × 120	6-crown 3-link
DREAMS-2G	WE31	Mg-Nd-Zn-Zr	PLLA	7	Sirolimus (1.4 µg/mm)	Cardiovascular	150 × 150	6-crown 2-link
UNITY-B™	MgNdMn21	Mg-Nd-Mn	/		/	Biliary	/	Y shaped

## Data Availability

No new data were created.

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
