# Peer review of "Research on the Current Application Status of Magnesium Metal Stents in Human Luminal Cavities"

_jfb, 2023, doi:10.3390/jfb14090462_

Round 1
Reviewer 1 Report
Biodegradable stent implantation is a promising procedure devoted to the remediation of luminal diseases. As an alternative to biodegradable polymers, the bioresorbable metals and especially the Mg-based intracorporal implants should be carefully explored for successful clinical implementations. The forms and content of this manuscript fall within the scope of the Journal Functional Biomaterials spanning the biomaterial area.
With appropriate terminology and reasonable argumentation, the manuscript clearly shows the principal areas of Mg stenting that are concentratedly on the diagram of Fig 1. The manuscript abstract reflects the general issues of the paper adequately The literature cited is quite relevant to this study and the illustrations (the tables and the figures) are executed unambiguously and accurately with a coherent interpretation.
Please discuss the preferences and drawbacks of two contemporary approaches in clinical applications to compare biodegradable polymers (PLA, PHB, PCL) and Mg-based stents. It is worth specifying the rate of resorption and form-memory feature, succinctly.
The enumeration in Bibliography should be corrected to keep a quoting sequence, for example, the reference in the legend to Fig. 6 is denoted as 135, but after that, in the text, the number 91 (L369) is followed. Please pay attention and elucidate if you have arranged a separate numeration for all of the figures.
In Fig. 8, the authors are invited to disclose the sense of the red arrows which point out probably the view of the stent after exploitation (A) and two unknown white points (C).
After the minor correction, the manuscript has to be promoted for the following publishing performance.
Reviewer 2 Report
The human body is made up of several tubular structures with essential functions responsible for transporting food, liquids, waste, and other substances throughout the body. As far as functional materials are sought, the literature reviewed by the authors is consistent with what they mention in the title of their manuscript, for which I would recommend the manuscript as it covers the most important aspects of the use of biodegradable and magnesium stents.
The human body is made up of several tubular structures with essential functions responsible for transporting food, liquids, waste, and other substances throughout the body. As far as functional materials are sought, the literature reviewed by the authors is consistent with what they mention in the title of their manuscript, for which I would recommend the manuscript as it covers the most important aspects of the use of biodegradable and magnesium stents.
Reviewer 3 Report
In this manuscript, the authors provided “Research on the current application status of magnesium metal stents in human luminal cavities”, defining an overview of the performance criteria, current research progress, and future trends in the field of stents, encompassing vascular, coronary, biliary, tracheal, esophagus, urethral, and intestinal stents. This manuscript could be helpful for the development of corrosion-resistant and biocompatible biodegradable Mg alloy stents required in different luminal environments. I would like to recommend its publication in this journal after addressing the following recommendations:
1) The types of Mg alloys used for stents, their advantages and disadvantages, and their biological and physicochemical behavior at different pH values should be discussed in the manuscript.
2) A section concerning the influence of the released hydrogen gas on normal cells and the cytotoxicity of Mg stents is missing.
3) The structure of the stents and the essence of fabrication methods (including coating procedures) should also be revealed in the text.
4) Please, always specify the structure of the stents and phase composition of the alloys in the reviewed studies.
5) Some statements need to be referred to. For example: „In addition to these properties, Mg implants promote tissue healing through collagen synthesis, angiogenesis, cell proliferation, and anti-inflammatory effects. These mechanisms contribute to the overall tissue regeneration process.“
6) Permission for the Figures with cited references should be asked.
7) Overall, a more critical view of the reviewed papers is desirable;
8) Please, always highlight the input of your research groups in the appropriate sections.
9) The challenge and trends in magnesium metal stents can be further discussed.
None
Round 2
Reviewer 3 Report
The authors have carefully addressed the reviewer's recommendations. The paper may be published in its present form.
None